# Divergent Seasonal Reproductive Patterns in Syntopic Populations of Two Murine Species in Southern Spain, *Mus spretus* and *Apodemus sylvaticus*

**DOI:** 10.3390/ani11020243

**Published:** 2021-01-20

**Authors:** Diaa Massoud, Miguel Lao-Pérez, Esperanza Ortega, Miguel Burgos, Rafael Jiménez, Francisco J. Barrionuevo

**Affiliations:** 1Departamento de Genética e Instituto de Biotecnología, Lab. 127, Centro de Investigación Biomédica, Universidad de Granada, Avenida del Conocimiento S/N, 18016 Armilla, Granada, Spain; dfm00@fayoum.edu.eg (D.M.); mlao@ugr.es (M.L.-P.); mburgos@go.ugr.es (M.B.); fjbarrio@go.ugr.es (F.J.B.); 2Department of Zoology, Faculty of Science, Fayoum University, Gamma St., Keman Square, Fayoum 63514, Egypt; 3Departamento de Bioquímica y Biología Molecular III e Inmunología, Facultad de Medicina, Universidad de Granada, Avenida de la Investigación 11, 18071 Granada, Spain; esortega@ugr.es

**Keywords:** seasonal breeding, seasonal testis regression, Muridae, Iberian peninsula, *Apodemus sylvaticus*, wood mouse, *Mus spretus*, Algerian mouse

## Abstract

**Simple Summary:**

In temperate zones of the Earth, most species reproduce in seasons providing the most favourable environmental conditions. Producing gametes is expensive in energetical terms, so both males and females either reduce or abolish gametogenesis during the non-breeding period. We thoroughly studied the testes of sexually inactive males of two rodents, the wood mouse, *Apodemus sylvaticus,* and the Algerian mouse, *Mus spretus*, in southern Iberian peninsula. These populations are syntopic, that is, animals of the two species share their territories and resources, so one would expect them to show similar or identical seasonal reproduction patterns. Contrarily, we found that both species reproduce during most of the year, but wood mice stop breeding in the summer whereas Algerian mice do it in winter. These divergent seasonal breeding patterns imply that either very subtle animal features and/or environmental cues operate to determine reproduction timing and support the notion that multiple models of circannual reproduction patterns are possible for different populations of the same species, showing that the mechanisms controlling seasonal reproduction are in fact very plastic and fast evolving. Hence, small mammals probably have multiple ways available to get adapted to the unstable environmental conditions derived from the ongoing global climate change.

**Abstract:**

In most mammals with seasonal reproduction, males undergo testis regression during the non-breeding period. We performed a morphological, hormonal, functional, and molecular study of the testes of sexually inactive males of two species of murine rodents, the wood mouse, *Apodemus sylvaticus,* and the Algerian mouse, *Mus spretus*, in syntopic populations of southern Iberian peninsula. Both species reproduce during most of the year, but wood mice stop breeding in the summer whereas Algerian mice do it in winter. Sexually inactive males of *A. sylvaticus* show complete testis regression with reduced levels of serum testosterone and abnormal distribution of cell-adhesion molecules. Contrarily, inactive males of *M. spretus* maintain almost normal spermotogenesis despite a significant reduction of androgenic function. The lack of an evident explanation for the divergent seasonal breeding patterns found in southern populations of *A. sylvaticus* and *M. spretus*, compared with northern ones, implies that very subtle species/population-specific features and/or non-conspicuous environmental cues probably operate to determine their seasonal breeding pattern. These results also support the notion that multiple models of circannual testis variation are possible for different populations of the same species, showing that the mechanisms controlling seasonal reproduction are in fact very plastic and fast evolving.

## 1. Introduction

In temperate zones of the Earth most species concentrate the reproductive effort in those seasons with the most favourable environmental conditions. In seasonally breeding mammals, both female and male gonads undergo substantial changes during the transition between the breeding and the non-breeding periods [1]. Two main aspects of seasonal breeding are currently being investigated: (1) the mechanisms of cyclic gonadal regression and recrudescence (mainly in the males), and (2) the environmental cues controlling circannual rhythms of reproduction. The process of testis regression have only been studied in a few species to date and, therefore, it is not yet well understood. Moreover, most studies are incomplete as they focus on just one particular aspect of the entire process (apoptosis, cell proliferation, hormonal production, morphological changes, dynamics of cell adhesion molecules [2,3,4,5]). Comprehensive studies including most of these features have been performed in the bank vole, *Clethrionomys glareolus* [6,7,8], the Iberian mole, *Talpa occidentalis* [9,10], the long hairy armadillo, *Chaetophractus villosus* [11,12], the greater white-toothed shrew, *Crocidura russula* [13], the South American plains vizcacha, *Lagostomus maximus* [14], the Egyptian long-eared hedgehog, *Hemiechinus auritus* [15] and the Syrian hamster, *Mesocricetus auratus* [16,17,18,19]. Therefore, more seasonal breeding mammals must be investigated in order to identify the mechanisms of testis regression which are evolutionarily conserved. Two main cellular mechanisms of germ cell depletion have been described during testis regression in seasonally breeding mammals: apoptosis and desquamation (sloughing) [1]. For years, apoptosis was the only cellular process known to be involved in seasonal testis regression, and it is the main cause of germ cell depletion in several mammalian species. However, most studies have been performed in the Syrian hamster, which has become a model species in the field of seasonal reproduction [16,17,18,19,20]. More recently, an alternative mechanism based on germ cell desquamation was reported in *T. occidentalis* [9,10], *C. villosus* [11] and *H. auritus* [15]. In these species, germ cells are sloughed alive and discarded through the epididymis, a process accompanied by both decreased levels of serum testosterone and disturbed distribution of the cell-adhesion molecules in the seminiferous epithelium [1]. Regarding the environmental cues controlling seasonal reproduction, photoperiod is the only one widely supported by evidence. For years, photoperiod has been considered the most common cue involved in this process, but the number of species representing exceptions to this rule is growing. In most cases, although it was possible to establish that seasonal reproduction is photoperiod-independent, the effector cue could be identified only in a few species, including the Californian mouse, *Peromyscus californicus,* in which reproduction depends on water availability [21], the Merino sheep, *Ovis aries,* where testis function depends on food availability and social factors [22] and the musk shrew, *Suncus murinus*, a species in which nutritionally challenged females stop breeding as a consequence of complete loss of sexual receptivity [23]. Jiménez et al. [24] showed that moles in southern Iberian peninsula reproduce in winter, whereas those from central Europe and Great Britain do it in the spring and summer. Also, Massoud et al. [13] reported that whereas males in northern populations of C. russula in the Iberian peninsula undergo seasonal testis regression (deduced from data reported by López-Fuster et al. [25], those from southern populations have active testes throughout the year. These facts prompted Jiménez et al. [1] to conclude that a particular pattern of seasonal reproduction is attributable to particular populations of a given species, but not to the species itself.

The wood mouse, *Apodemus sylvaticus*, is a small rodent of the family Muridae and endemic from Europe and north-western Africa. This species frequently inhabits forest habitats but also steppes, crop fields, rocky and mountain environments [26].

It has been suggested that the reproductive cycle of *A. sylvaticus* in central and north Europe starts early in the spring and ends by late summer or early autumn [27,28]. By contrast, the annual breeding period for the wood mice in the Mediterranean scrubland of south-western Spain is extended from August to April [28].

The Algerian mouse, also called the western Mediterranean mouse, *Mus spretus*, is a myomorphic rodent species also belonging to the family Muridae. This mammal inhabits south-western Europe and north-western Africa [29,30]. It prefers open terrain, avoiding dense forests. The circannual reproductive cycle of *M. spretus* in southern Iberian peninsula was studied by Vargas et al. [31]. Two well-marked phases were defined for this species: a period of sexual inactivity in winter extending from November to January, with a reduction in the size of testes and seminal vesicles, and a long period of sexual activity during the rest of the year.

No study on the process of seasonal testis regression has been performed in these two species, and this is the aim of this work. We performed a comprehensive study of the testes in populations of both *A. sylvaticus* and *M. spretus* during the two periods (sexually active and inactive) of their respective reproductive cycles in south-eastern Iberian peninsula, including histological and immunohistological analyses (detection of both somatic and germ cell markers), morphometry (body and testis mass, seminiferous tubules diameter), blood-testis barrier (BTB) permeability, serum testosterone levels and the incidence of apoptosis. Interestingly, we found a divergent pattern of seasonal reproduction in these two species, despite the fact that they live syntopically in the studied populations.

## 2. Materials and Methods

### 2.1. Animals and Tissue Collection

Seventeen adult males of the wood mouse *Apodemus sylvaticus* were captured alive in small isolated populations near the localities of Las Gabias and La Malahá (Granada province, south-eastern Spain) using both medium-sized Hipólito type traps and Havahart SHA1020 two-door mouse traps, baited with pieces of apple and bread with crude olive oil. Two study groups were established: (1) animals captured from September to May (breeding period, *n* = 10) and (2) those captured from June to August (non-breeding period, *n* = 7; Appendix A). In addition, twenty-eight adult males of the Algerian mouse *Mus spretus* were captured using plastic trip-traps (NHBS ref. 176705) baited as described above and provided with dry hay to keep the animals warm. Two study groups were established: (1) animals captured from March to November (breeding period, *n* = 21) and (2) those captured from December to February (non-breeding period, *n* = 7; Appendix A). The animals were captured in the daytime during the winter and at night during the summer, with permission of the Andalusian Environmental Council (Consejería de Agricultura, Pesca y Medio Ambiente). Animals were handled following the guidelines and approval of the Ethical Committee for Animal Experimentation of the University of Granada (Registration number: 450-19131) and were euthanised by cervical dislocation or by CO_2_ inhalation. Animals were dissected and the gonads removed, weighed, and fixed overnight in a 50× volume of Serra’s fixative (100% ethanol, 40% formaldehyde, and glacial acetic acid in proportions 60:30:10). Epididymides were also processed for histological studies. The three regions (caput, corpus and cauda) were analysed. Further analyses and experiments (protein immunodetection, BTB permeability and apoptosis incidence) were performed using the testes from several (4–6) males showing the highest body and testis mass in each period.

### 2.2. Histology and Immunohistological Methods

Testes and epididymides were embedded in paraffin, sectioned (5 μm), mounted on polylysine-coated slides (VWR, Leuven, Belgium), and stained with haematoxylin and eosin according to standard procedures for morphological analysis. Immunohistochemistry was performed only on testis sections. For this, we used the ABC Kit (Vector Laboratories, Burlingame, CA, USA), according to the manufacturer’s instructions. For single and double immunofluorescence, preparations were incubated overnight with primary antibodies, washed, incubated with the appropriate conjugated secondary antibodies for 1 h at room temperature and treated with 4′,6-diamino-2-phenylindol (DAPI). Digital photomicrographs were taken in a Nikon Eclipse Ti microscope (Nikon Corporation, Tokio, Japan). Three testicular sections from each of the groups of males of both species were mounted on the same slide and processed together. The primary antibody was omitted in negative controls. Appendix A summarises the antibodies and working concentrations used in this study.

### 2.3. Apoptosis

Apoptotic cells present in testicular histological preparations were revealed by the TUNEL method using the *Fluorescent* In Situ *Cell Death Detection Kit* (Roche ref. 11684795910, Roche, Mannheim, Germany), according to the manufacturer’s instructions. The enzyme solution was omitted in negative controls.

### 2.4. Serum Testosterone Levels

Blood samples were collected from males, stored at 4 °C overnight and centrifuged at 6000 rpm for 20 min at 4 °C. The supernatant was collected and stored at −80 °C until used. Hormone concentrations were measured by radioimmunoassay (RIA) using the DRG Testosterone RIA (CT) Kit (DRG, New York, NY, USA), according to standard procedures. Duplicate measurements were made for all animals. The analytical sensitivity of the kit was 0.05 ng/mL, the inter-assay coefficient of variation was 4.8%, and the intra-assay coefficient of variation was 3.3%. The specificity, determined as the percentage of cross-reaction estimated by comparing the concentration yielding a 50% inhibition was 0.31% for dihydrotestosterone and 0.28% for androstenedione.

### 2.5. Blood-Testis Barrier Permeability Assay

To test the permeability of the blood-testis barrier (BTB), two males of each species captured in both the breeding and the non-breeding seasons were anaesthetised with 0.125% avertin (2,2,2-tribromoethanol), their testes were exposed, and a total of 50 μL of 10 mg/mL EZ-Link Sulfo-NHS-LC-Biotin (Pierce Chemical Co., Rockford, IL, USA) diluted in PBS containing 1 mM CaCl_2_ was injected beneath the tunica albuginea of the left testis. The right testis was injected with PBS with 1 mM CaCl_2_ as a control. The testes were placed again inside the scrotal sac, and the animals were euthanised 30 min later. The testes were immediately removed and fixed overnight in 4% paraformaldehyde, dehydrated, embedded in paraffin, sectioned and mounted on slides. After deparaffination and rehydration, the tracer was detected by incubating the sections for 30 min with an Alexa Fluor 568-conjugated streptavidin solution (included in the tracer kit, Pierce Chemical Co., Rockford, IL, USA) at 25 °C and treated with DAPI.

### 2.6. Morphometrics and Statistics

The diameter of at least 10 seminiferous tubules of the same testis section was measured in three animals of each species captured in the two seasons of their respective reproductive cycle. We used the manual measurement tool of the NIS-Elements Advanced Research (AR) application that controls a Nikon DS-Fi1c digital camera (Nikon Corporation, Tokio, Japan). Seminiferous tubule diameter (microns), and body (g) and testis mass (mg) are reported as mean ± standard deviation values. As data fit a normal distribution, we used Student’s *t*-tests for mean comparisons. For apoptotic cell counting, the total number of apoptotic cells present in at least five 10× photomicrographs (0.55 mm^2^) of three animals of each species and each season was recorded. Given the particular distribution of the apoptotic cells, these data do not fit a normal, but a negative binomial distribution. Accordingly, we used the non-parametric Wilcox’s test for mean comparison. We used the R packages stats [32] and dplyr [33] to perform the statistical tests.

## 3. Results

### 3.1. Sexually Inactive Males Undergo Testis Regression in Apodemus sylvaticus but Not in Mus spretus

The body mass of the *A. Sylvaticus* males analysed in the present study showed no difference between sexually active (non-summer) and inactive (summer) males (Figure 1A; Appendix A). In contrast, both testis mass and relative mass of the testis (expressed as testis mass (mg)/body mass (mg) × 100) revealed a significant 65% reduction in inactive males as compared with active ones (Figure 1B,C; Appendix A), showing that in this population reproduction occurs throughout the year except in summer. Regarding *M. spretus* we found no difference between the body mass of active (non-winter) and inactive (winter) males (Figure 1A; Appendix A). In contrast, we found a reduction of about 28% of both the testis mass and the relative mass of the testis in inactive males when compared with active ones (Figure 1B,C; Appendix A).

The histological analysis showed that active males of both *A. sylvaticus* and *M. spretus* had fully functional testes with normal seminiferous tubules showing all stages of the spermatogenic cycle and complete spermiogenesis (Figure 1E,I). Accordingly, we found abundant sperm in their epididymides (Figure 1F,J). In contrast, inactive males showed significantly reduced diameter of the seminiferous tubules in *A. sylvaticus* and 15% in *M. spretus*; Figure 1D; Appendix A). However, whereas the seminiferous tubules of the inactive males of *A. sylvaticus* showed no recognisable stage of the spermatogenic cycle, most of them containing only primary spermatocytes in the adluminal compartment and thus lacking secondary spermatocytes, spermatids, and sperm (Figure 1G), those of *M. spretus* appeared histologically normal and fully functional (Figure 1K). Consistently, the diameter of the epididymal tubules of the inactive males of *A. sylvaticus* was reduced and empty (Figure 1H) whereas that of *M. spretus* inactive males appeared full of sperm (Figure 1L).

### 3.2. Meiosis Is Arrested in the Inactive Testes of A. sylvaticus but Not in Those of M. spretus

We analysed the expression pattern of several testicular cell markers in both sexually active and inactive males of both species. Previous studies have shown that expression of SOX9, a Sertoli cell marker, is spermatogenic stage-dependent in adult testes of both the rat and the Iberian mole, being stronger at stages I–IV [34,35]. Immunohistochemistry for SOX9 in active testes of both species as well as in inactive testes of *M. spretus* revealed the same expression pattern as that reported for rats and moles (Figure 2A,C,D). By contrast, inactive testes of *A. sylvaticus* exhibited a uniform SOX9 staining intensity and very reduced separation between the nuclei of contiguous SOX9-positive cells, as a consequence of the strong shrinkage that these seminiferous tubules undergo (Figure 2B). Double immunofluorescence for SOX9 and DMRT1, which is a marker for both Sertoli and spermatogonial cells [36], also showed that only inactive testes of *A. sylvaticus* presented an altered DMRT1 expression pattern in Sertoli cells (Figure 2E–H, yelow nuclei). Next, we analysed the expression pattern of germ cell markers. As mentioned above, DMRT1 is also expressed in spermatogonia (Figure 2E–H, red nuclei). This expression is spermatogenic stage-dependent, existing tubules with a high number of DMRT1^+^ spermatogonial cells (stages VI–VII) and others containing a low number of them (stages II-IV [35]). Double immunofluorescence for SOX9 and DMRT1 revealed that active males of the two species as well as inactive males of *M. spretus* presented such an expression pattern (Figure 2E,G,H, red cells). However, the inactive testes of *A. sylvaticus* contained two types of tubules with different amounts of DMRT1^+^ cells, including tubules in which DMRT1 expression is almost exclusively restricted to Sertoli cells (Figure 2F, stars) and those in which the DMRT1 signal is also found in spermatogonial cells located in the basal compartment of the germinative epithelium (Figure 2F, asterisks). These observations show that inactive testes experience the spermatogonial proliferative phases of the spermatogenic cycle. We also studied by immunofluorescence the expression of DMC1, a marker for zygotene and early pachytene spermatocytes [37]. In active males of the two species as well as inactive males of *M. spretus* we found a spermatogenic stage-dependent expression of DMC1, which was only seen at stages VII-IX (Figure 2I,K,L). In inactive testes of *A. sylvaticus* DMC1 was also observed in some testis tubules but not in others (Figure 2J). Finally, DMRT1-DMC1 double immunofluorescence showed no colocalisation of both proteins in any cell of either active or inactive testes of both species (Figure 2M–P). Altogether, these observations show that in inactive testes of *A. sylvaticus* both spermatogonial mitosis and meiosis entry were maintained and thus the early stages of spermatogenesis remained active, but spermatocytes did not progress beyond the pachytene stage.

### 3.3. The Blood Testis Barrier Is Impaired in the Inactive Testes of A. sylvaticus but Not in Those of M. spretus

To check the status of the blood testis barrier (BTB), we performed immunohistochemistry for claudin 11 (CLDN11), a principal component of the tight junctions forming the BTB [38]. In the active testes of *A. sylvaticus* and in the both active and inactive testes of *M. spretus,* we found a sharp CLDN11 immunoreactivity surrounding the testis tubules just beneath the basal lamina (Figure 3A,C,D). In contrast, CLDN11 expression was completely disorganised in the inactive testes of *A. sylvaticus* (Figure 3B). We next tested the integrity of the BTB by injecting a biotin tracer into the interstitial space of the testes. In active males of *A. sylvaticus* and in all males of *M. spretus*, the biotin tracer was detected in the interstitial tissue as well as in the basal compartment of the seminiferous tubules, but it did not reach the adluminal compartment (Figure 3E,G,H). In contrast, biotin immunoreactivity was clearly observed in both basal and adluminal compartments of the testes from inactive males of *A. sylvaticus*, evidencing that testis regression includes BTB permeation in these animals (Figure 3F). Consistent with these results, CLDN11-DMC1 double immunofluorescence showed that the CLDN11 expression domain was always interposed between the DMC1^+^ cells and the basal lamina in the active males of *A. sylvaticus* and in all males of *M. spretus* (Figure 3I,K,L), showing that all the cells passing through the BTB are early meiocytes. Contrarily, the testes in inactive males of *A. sylvaticus* showed patches of CLDN11 expression ectopically located in the adluminal compartment, more inner than some DMC1^+^ cells, indicating that permeation is caused by disruption of the BTB components (Figure 3J). We also checked the status of the *lamina propria* in active and inactive males of both species using immunohistochemistry for ACTA2, DESMIN, and LAMININ. All three proteins were detected around the seminiferous tubules, in both active and inactive testes, suggesting that the *lamina propria* was not altered during testis regression in these species (Appendix A).

### 3.4. Androgen Levels Are Reduced in the Inactive Males of Both A. sylvaticus and M. spretus

To study the androgenic function in active and inactive males of both *A. sylvaticus* and *M. spretus,* we initially analysed by immunohistochemistry the expression patterns of the androgen receptor (AR) and the cholesterol side-chain cleavage enzyme (cytochrome P450scc). AR expression was seen in Sertoli, Leydig, and peritubular myoid cells in the testes of both active and inactive males of the two species, showing a similar immunoreactivity (Figure 4A–D). Likewise, P450scc was strongly expressed in Leydig cells, exhibiting similar signal intensity in all mice of either species (Figure 4E–H). Finally, we also determined the serum concentration of testosterone by radioimmunoassay (RIA). For *A. Sylvaticus,* we found a significant 70% reduction in serum testosterone concentration in inactive males compared to active ones (active: 0.96 ± 0.69 ng/mL, *n* = 6; Inactive: 0.29 ± 0.23 ng/mL, *n* = 6; two-tailed *t*-test, *p* = 0.048; Figure 4I). In the case of *M. spretus,* the reduction in inactive males was 80% of the levels measured in active ones (active: 0.065 ± 0.043 ng/mL; inactive: 0.013 ± 0.007 ng/mL; *n* = 6; two-tailed *t*-test, *p* = 0.011; Figure 4I).

### 3.5. Increased Apoptosis in Inactive Testes of A. sylvaticus

We performed terminal deoxynucleotidyl transferase dUTP nick end labeling (TUNEL) assays to check whether apoptosis is altered in the testes of *A. sylvaticus* and *M. spretus* during their circannual reproductive cycles. In the testes of active males of *A. sylvaticus* and in those of *M. spretus* at any season, most of the seminiferous tubule sections were devoid of apoptotic cells (Figure 5A,C,D). In contrast, we found many apoptotic cells within the seminiferous tubules of the inactive testes of *A. sylvaticus* (Figure 5B). The difference in the number of apoptotic cells between active and inactive testes of this species was highly significant (active: 5.2 ± 10.7 cells/0.5 mm^2^; inactive: 169.9 ± 38.4 cells/0.5 mm^2^; Wilcox test, *p* = 1.03 × 10^−10^; Figure 5E). However, in *M. spretus*, we found no significant difference between active and inactive males (active: 6.8 ± 7.6 cells/0.5 mm^2^; inactive: 5.1 ± 5.5 cells/0.5 mm^2^; Wilcox test, *p* = 0.29; Figure 5E).

## 4. Discussion

The two rodent species studied here belong not only to the same family (Muridae), but also to the same subfamily (Murinae) and their populations in southern Iberia are not only sympatric, but also syntopic, occupying the same habitat. In fact, we captured animals of both species in the same locations, in scrublands and water streams banks, setting their respective traps mixed and separated by just a few meters from each other. Despite this quite close phylogenetic relationship and similar lifestyles, the wood mice stop breeding in summer and the Algerian mice do it in winter, showing divergent seasonal reproductive patterns. However, the surprise here is not the breeding pattern of *A. sylvaticus*, but that of *M. spretus*. In most European species of small mammals, the populations of southern Iberian peninsula show inverted breeding pattern when compared with those located in central and northern Europe, or even with those of northern Iberia. This is the case for *A. sylvaticus*, a species whose populations in northern Europe breed between spring and autumn, with a period of latency in winter [39], whereas southern populations breed throughout the year except for a dormancy period in summer ([40] and present paper). The same phenomenon has been described in the Iberian mole (*Talpa occidentalis)* [9,10,24], the greater white toothed shrew (*Crocidura russula)* [13], the Mediterranean pine vole (*Microtus duodecimcostatus*), the water vole (*Arvicola sapidus*) (our unpublished data), and probably many other species whose populations find the worst living conditions in northern winters and in southern summers. This fact supports the widely accepted notion that small mammals with a short lifespan, including many rodents, are mainly opportunistic breeders in which various environmental cues such as food and water availability and/or temperature, rather than photoperiod, are the main factors determining their energy balance and thus their breeding pattern [41]. This is especially true for the lower latitudes of the temperate zone of the Earth [42], where most of the small mammals studied are insensitive to photoperiod variations (e.g., the white-footed mouse, *Peromyscus leucopus*, in south-eastern areas of the Nearctic zone [43]). To our knowledge, *M. spretus* is the only known small mammal species not showing such an inverted breeding pattern in European southern populations with respect to northern ones. The cause could be a particular higher sensitivity of these tiny animals to low temperatures. In fact, Vargas et al. [31] suggested that circannual variations in the duration of the reproductive cycle of southern Iberian populations of this species seem to be mainly due to environmental conditions, particularly temperature. Also, whereas other small murine rodents like *A. Sylvaticus* or *Mus musculus* (and many others) inhabit central and northern regions of the paleartic zone, the distribution area of *M. spretus* is restricted to southern areas (Spain and Portugal, excluding the Cantabrian mountain range, south of France and large regions of northern Africa). Although it has been suggested that food availability can affect the reproductive activity of *A. sylvaticus* [44], both species can feed on a wide variety of seeds, fruits, and insects [45,46,47,48], so food availability seems not to be the direct cause of their breeding pattern divergence. In fact, they both overlap their respective breeding periods in spring and autumn. Nevertheless, if we take into account that summer and winter offer the less favourable life conditions due to extreme temperatures, the current situation could represent an adaptive advantage as breeding pattern divergence reduces the competence for scarce resources during summer and winter.

The magnitude of the alterations that testes of *A. sylvaticus* and *M. spretus* undergo during their respective non-breeding periods is also divergent, as males of *M. spretus* exhibit a moderate although significant testis reduction in winter, whereas those of *A. sylvaticus* experience complete testis regression in summer. The situation in the latter species, that is, complete testis regression, is the most frequent among the seasonal breeding species investigated to date. However, the lack of testis regression even when reproduction is halted has only been described in populations of the great white toothed shrew, *Crocidura russula,* located in the same region where mice studied in the present paper were captured [13]. We proposed that the lack of testis regression in the shrew is a new adaptive mechanism derived from two particular features of the shrews: (1) males of this species have small testes, so the investment in spermatogenesis is low [49,50]; (2) the spermatogenic cycle of this species is slow and long. Thus, if the non-breeding period is short, as it is expected to occur in southern populations as shown in talpid moles [24], then testis regression would not be efficient in terms of energy-saving because maintaining small testes is not onerous and recovering from testis regression would be too slow. The length of the spermatogenic cycle in *M. spretus* is currently unknown, but our data show that, compared with *A. sylvaticus*, the relative testis mass of *M. spretus* is three times smaller. Hence, the situation in the latter species could be similar to that described for *C. russula*. Another important factor could be the very short life expectancy of *M. spretus,* which is of only four months in the wild [51]. In these circumstances, becoming sterile for near two months (the duration of spermatogenic cycle is about 34.5 days in mice [52]) is not affordable for those animals that have to pass through a non-breeding period as there is a high probability that they will not survive the time needed to activate the spermatogenic cycle again when the females start to be receptive at the beginning of the next breeding period. This is not the case of *A. sylvaticus,* whose life expectancy is of about 12 months in the wild [29].

Algerian mice captured during the non-breeding season show normal spermatogenesis and spermiogenesis despite the fact that they undergo a reduction in testicular mass and seminiferous tubule diameter. Also, their epididymides contained mature spermatozoa, indicating that they are fully fertile. However, they are probably sexually inactive due to (1) the low levels of serum testosterone and (2) the lack of sexual receptivity of the females. The lower testis mass of *M. spretus* in the non-breading period is clearly a consequence of a reduced diameter of the seminiferous tubules (around 15% shorter than that of active males). This reduction could originate either from a lower number of spermatogonial cells entering meiosis (spermatogonial proliferation slows down) or by increased levels of primary spermatocyte depletion by apoptosis. Our results clearly support the former possibility, as we found no significant difference in the number of apoptotic cells between testes from active and inactive males.

The situation in the regressed testes of *A. sylvaticus* was completely different. In this case, spermatogonial mitosis and meiosis entry were not interrupted, but meiotic germ cells die by apoptosis around the zygotene-pachytene stage. We previously reported a similar situation in the regressed testes of the Iberian mole, *T. occidentalis* [9]. It is possible that, like in the Iberian mole, the role of apoptosis in *A. sylvaticus* is to eliminate primary spermatocytes in the inactive testes, but it does not contribute substantially to the massive germ-cell depletion occurring during the process of testis inactivation. Apoptosis, rather than being the cause of testis regression, could be the cellular process, ensuring that seminiferous tubules maintain their regressed status during the non-breeding period. Further analyses of testes from males captured at the precise time of testis regression will help to elucidate this question. Nevertheless, our current results suggest a mechanism of testis regression in *A. sylvaticus* based on germ cell desquamation (sloughing), similar to that we also described in the Iberian mole [10]. In this species, seasonal testis regression is mediated by desquamation of living germ cells due to disorganisation of the cell junctions that maintain the architecture of the germinative epithelium. Sertoli cells establish several types of cell junctions with adjacent Sertoli cells, thus forming the BTB, as well as with pre-meiotic, meiotic and post-meiotic germ cells [53,54,55,56]. A desquamation-based germ cell depletion requires the disruption of the cell-adhesion molecules forming these cell junctions. Here we report evidence that testis regression in the wood mouse includes BTB permeation caused by disruption of the BTB proteins, including CLDN11, suggesting that germ cell depletion could also be caused by disruption of the cell adhesion molecules forming the junctions between Sertoli and germ cells. Again, we need to study testes from males undergoing the process of testis regression to test this hypothesis.

Both, increased apoptosis and BTB permeation in the regressed testes of *A. sylvaticus* can be explained by the reduction of testosterone levels we observed in inactive males. It is well known that (1) intra-testicular androgens, whose concentration correlates well with that in the serum, negatively regulate testicular apoptosis [57,58,59], and (2) that testicular cell-junctions are regulated by hormones, although the underlying mechanisms are not yet completely understood. In this context, Sertoli cell-specific ablation of the androgen receptor gene (*AR*) showed that testosterone regulates BTB permeability by controlling the expression of *Cldn3* (the gene for CLDN 3) and that the subsequent BTB permeation leads to the loss of the testicular immune privilege [60]. Furthermore, it was reported that testosterone upregulates CLDN11 expression in primary cultures of mouse and rat Sertoli cells [61,62], whereas both follicle-stimulating hormone and testosterone regulate the expression and location of CLDN11 and the subsequent formation of tight junctions between rat Sertoli cells cultured in vitro [63]. Thus, the question arises as to why low levels of serum testosterone in inactive males of *M. spretus* induce neither increased apoptosis nor both CLDN11 expression and BTB disruption. Several explanations are possible for this question: (1) the effects of testosterone on these processes may not be exactly the same in all species, (2) the intra-testicular levels of testosterone may be finely controlled, existing different effects induced at different testosterone reductions, and (3) additional control elements, other than androgens, are likely to exist that may establish differences between the Algerian mouse and other seasonal breeders. Perhaps these differences are the adaptive response of southern populations of this species to their need of avoiding complete testis regression.

Our finding of a divergent seasonal breeding pattern in southern populations of *A. sylvaticus* and *M. spretus* implies that either very subtle species- and/or population-specific features or non-conspicuous environmental cues, or both, may operate to define a particular circannual reproduction timing. The genetic and physiological mechanisms by which apparently irrelevant environmental cues are transduced to induce a reproductive response remain obscure, and studies like this may help to unravel them. In the light of current knowledge, two conclusions can be drawn regarding the mechanisms controlling seasonal reproduction: (1) that they are in fact very plastic and much less rigid than initially considered, and (2) that they appear to be fast-evolving. Hence, mammalian populations (at least those of small species) probably have available multiple ways to get adapted to the unstable environmental conditions derived from ongoing global climate change.

## 5. Conclusions

In southern Iberian peninsula, syntopic populations of two species of murine rodents, the wood mouse, *Apodemus sylvaticus,* and the Algerian mouse, *Mus spretus*, show divergent seasonal breeding patterns. Both species reproduce during most of the year, but wood mice stop breeding in the summer whereas Algerian mice do it in winter. Sexually inactive males of *A. Sylvaticus* show complete testis regression mediated by reduced levels of serum testosterone and abnormal allocation of cell-adhesion molecules. This includes spermatogenic arrest and blood-testis barrier permeation. The role of apoptosis in these regressed testes is to eliminate meiotic germ cells. Contrarily, sexually inactive males of *M. spretus* maintain almost normal spermatogenic activity and intact blood-testis barrier despite a significant reduction of the androgenic function. The lack of an evident explanation for our finding of a divergent seasonal breeding pattern in southern populations of *A. sylvaticus* and *M. spretus* implies that either very subtle species/population-specific features and/or non-conspicuous environmental cues probably operate to determine the seasonal breeding pattern. Our results also support the notion that multiple models of circannual testis variation may exist for different populations of the same species, showing that the mechanisms controlling seasonal reproduction are in fact very plastic and fast-evolving.

## Figures and Tables

**Figure 1 animals-11-00243-f001:**
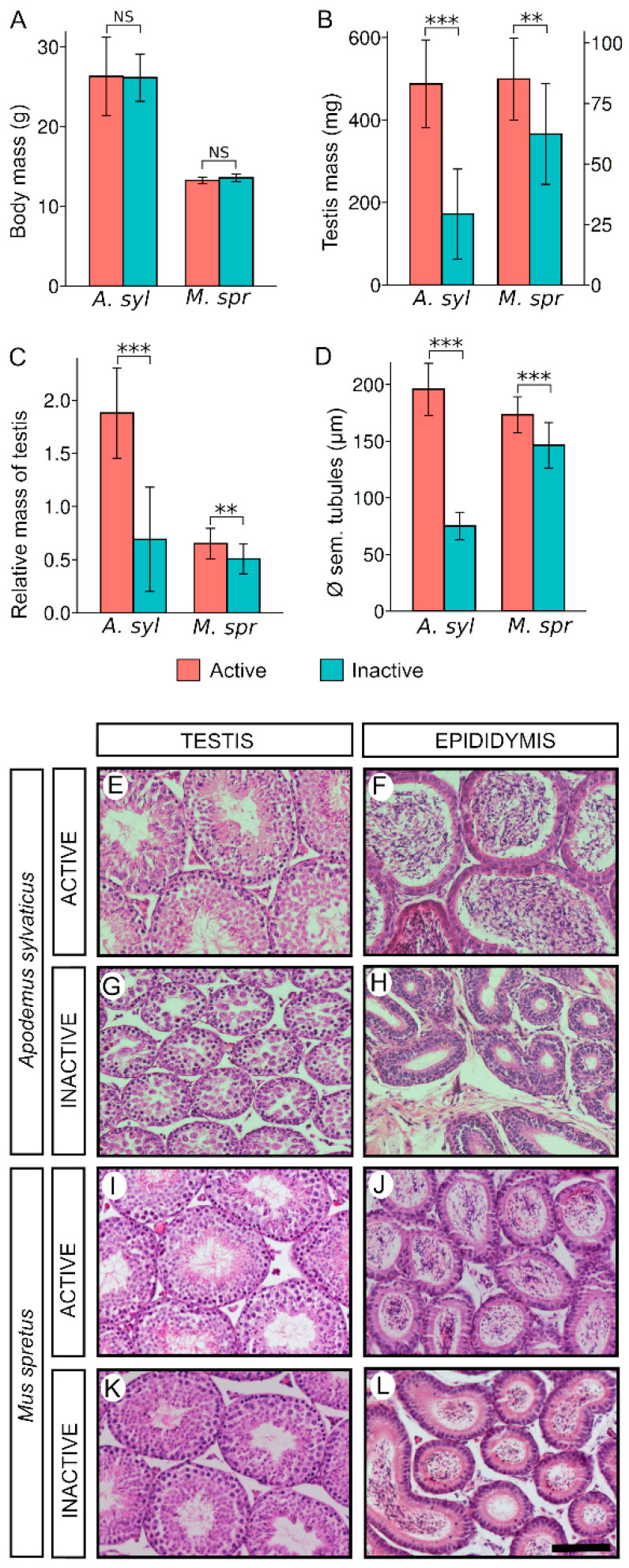
Morphometric analysis and histology of testes and epididymes of *Apodemus sylvaticus* and *Mus spretus* during the reproductive cycle. (**A**–**D**) Comparisons of (**A**) body mass, (**B**) testis mass, (**C**) relative mass of the testis and (**D**) seminiferous tubule diameter of *A. sylvaticus* and *M. spretus* during their reproductive cycle. Asterisks indicate the level of statistical significance (**: *p* < 0.01; ***: *p* < 0.001); NS: not significant (**E**–**L**) Hematoxylin-eosin-stained histological sections of testes (**E**,**G**,**I**,**K**) and epididymides (**F**,**H**,**J**,**L**) from *A. sylvaticus* (**E**–**H**) and *M. spretus* (**I**–**L**) belonging to either active (**E**,**F**,**I**,**J**) or inactive (**G**,**H**,**K**,**L**) study groups. Scale bar shown in (**L**) represents 100 μm for all pictures.

**Figure 2 animals-11-00243-f002:**
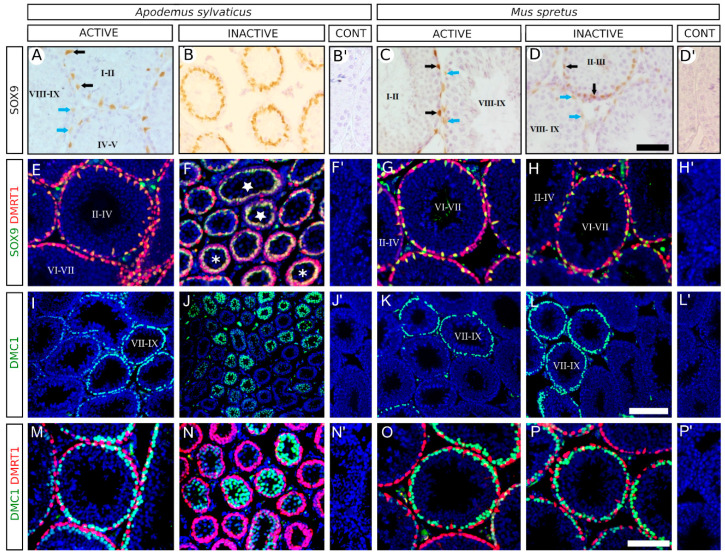
Immunohistological analysis for several Sertoli and germ cell-specific molecular markers in testis sections from *A. sylvaticus* and *M. spretus*. (**A**–**D**) Immunohistochemistry for SOX9. In active testis of *A. sylvaticus* (**A**) as well as in active and inactive testes of *M. spretus* (**C**,**D**) SOX9 is expressed in Sertoli cells with a stronger expression in stages I–IV (black arrows) and a weaker expression in stages VII–X (blue arrows). In contrast, in inactive testes of *A. sylvaticus* (**B**) SOX9 present a strong uniform staining in all the Sertoli cells. (**E**–**H**) Double immunofluorescence for SOX9 (green) and DMRT1 (red). DMRT1 is expressed in spermatogonial cells (red cells) and in Sertoli cells, where co-localises with SOX9 (yellow cells). In active testes of *A. sylvaticus* (**E**) and active and inactive testes of *M. spretus* (**G**,**H**) DMRT1 present a spermatogenic cycle-dependent expression, with tubules containing a higher number of DMRT1-expressing spermatogonial cells corresponding to stages VI–VII and with tubules containing a lower number of DMRT1-expressing spermatogonial cells corresponding to stages II–IV. In inactive testes of *A. sylvaticus* (**F**) also is visible two types of tubules with different amount of DMRT1-positive cells, including tubules in which DMRT1 expression is almost exclusively restricted to Sertoli cells (stars) and tubules in which the DMRT1 signal is also found in spermatogonial cells located in the basal compartment of the germinative epithelium (asterisks). (**I**–**L**) Immunofluorescence for DMC1. In the active testes of *A. sylvaticus* (**I**) and in the active and inactive testes of *M. spretus* (**K**–**L**) DMC1 is expressed in testis tubules in the spermatogenic-stages VII–IX. In inactive testes of *A. sylvaticus* DMC1 is also expressed in some testis tubules but not in others (**J**). (**M**–**P**) Double Immunofluorescence for DMC1 and DMRT1. In both active and inactive testes of *A. sylvaticus* (**M**,**N**) and *M. spretus* (**O**,**P**) both proteins never co-localised. Testis sections in (**E**–**P**) were counterstained with 4′,6-diamino-2-phenylindol (DAPI). Inserts in (**B’**,**D’**,**F’**,**H’**,**J’**,**L’**,**N’**,**P’**) show negative controls. Scale bar shown in (**D**) represents 50 μm for (**A**–**D**). Scale bar shown in (**L**) represents 200 μm for (**I**–**L**). Scale bar shown in (**P**) represents 100 μm for (**E**–**H**) and (**M**–**P**).

**Figure 3 animals-11-00243-f003:**
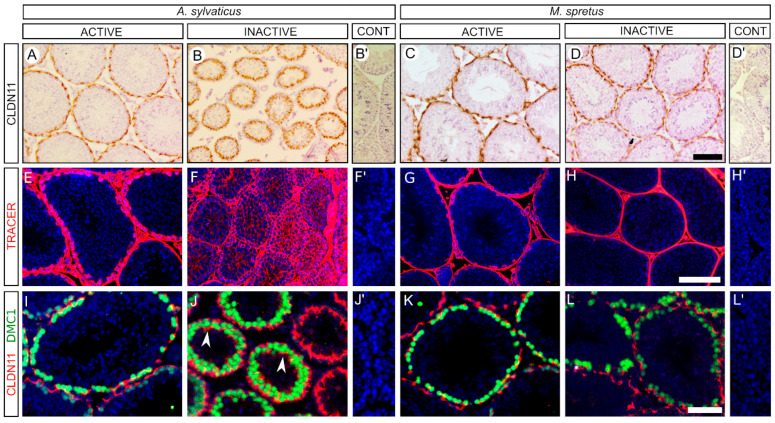
The blood-testis barrier BTB *of Apdemus sylvaticus* but not of *Mus spretus* is impaired in the non-breeding season. (**A**–**D**) Immunohistochemistry for CLAUDIN11. In the active testes of *A. sylvaticus* (**A**) as well as in the active and inactive testes of *M. spretus* (**C**,**D**) a sharp CLAUDIN11 immunoreactivity in the basal part of testis tubules could be observed. In contrast, Claudin11 expression is completely disorganised in the inactive testes of *A. sylvaticus* (**B**) (**E**–**H**) Test of BTB functionality using a biotin tracer (red fluorescence). In the active testes of *A. sylvaticus* (**E**) and the active and inactive testes of *M. spretus* (**G**,**H**), the tracer did not penetrate beyond the basal compartment, whereas in inactive testes of *A. sylvaticus* (**F**) it reached the deepest areas of the regressed seminiferous tubules. (**I**–**L**) Double immunofluorescence for DMC1 (green), and for CLAUDIN 11 (red). In the active testes of *A. sylvaticus* (**I**) and in the active and inactive testes of *M. spretus* (**K**,**L**) DMC1-positive germ cells are inner to CLAUDIN 11 expression domain, whereas in inactive testes of *A. sylvaticus* (**J**) CLAUDIN 11 expression (arrowheads) could be observed inner to DMC1-positive germ cells. Testis sections in (**E**–**L**) were counterstained with DAPI. Inserts in (**B’**,**D’**,**F’**,**H’**,**J’**,**L’**) show negative controls. Scale bar shown in (**D**) represents 100 μm for (**A**–**D**). Scale bar shown in H represents 100 μm for (**E**–**H**). Scale bar shown in (**L**) represents 50 μm for (**I**–**L**).

**Figure 4 animals-11-00243-f004:**
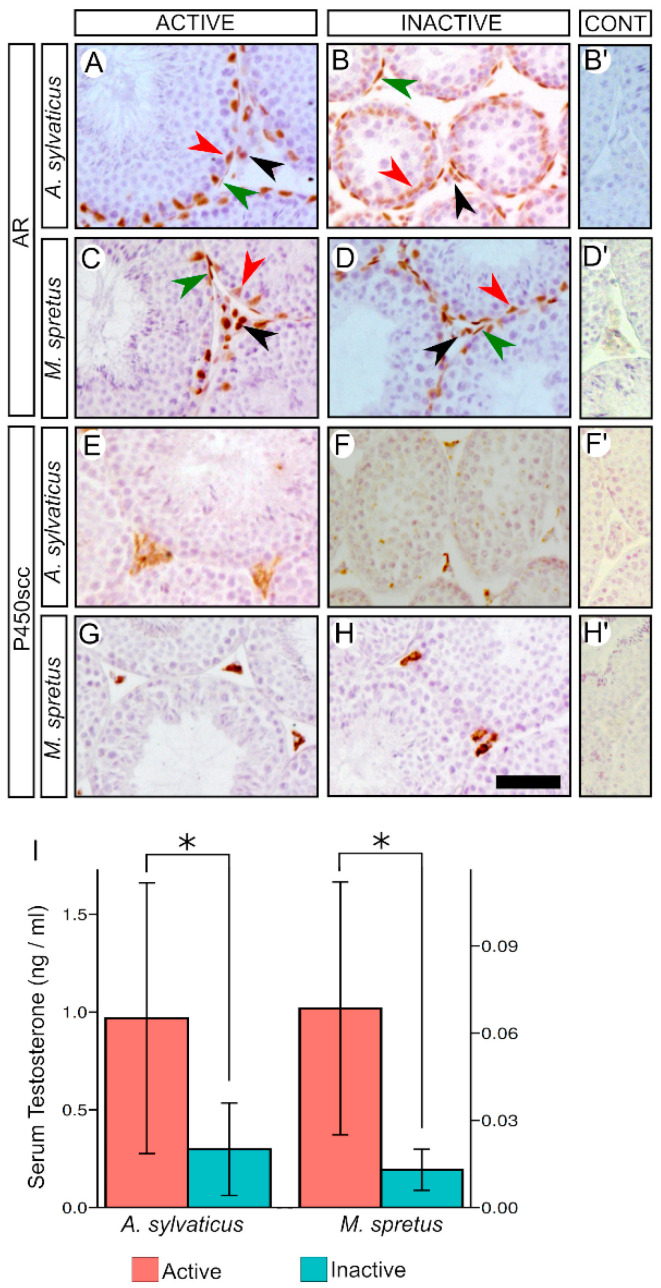
Reduced steroidogenesis during the non-breeding season of *Apodemus sylvaticus* and *Mus spretus*. (**A**–**H**) Immunohistochemistry for androgen receptor (AR) and cholesterol side-chain cleavage enzyme (P450scc). AR was expressed in Sertoli (red arrowhead), Leydig (black arrowhead), and peritubular myoid cells (green arrowhead) in active and inactive testes of both species with a similar immunoreactivity (**A**–**D**). P450scc was strongly expressed in Leydig cells with similar signal intensity between active and inactive testes of both species (**E**–**H**). (**I**) Serum testosterone concentrations in active and inactive males of *A. sylvaticus* and *M. spretus*. Inserts in (**B’**,**D’**,**F’**,**H’**) show negative controls. Scale bar shown in (**H**) represents 50 μm for (**A**–**H**). Asterisk indicate the level of statistical significance (*: *p* < 0.05)

**Figure 5 animals-11-00243-f005:**
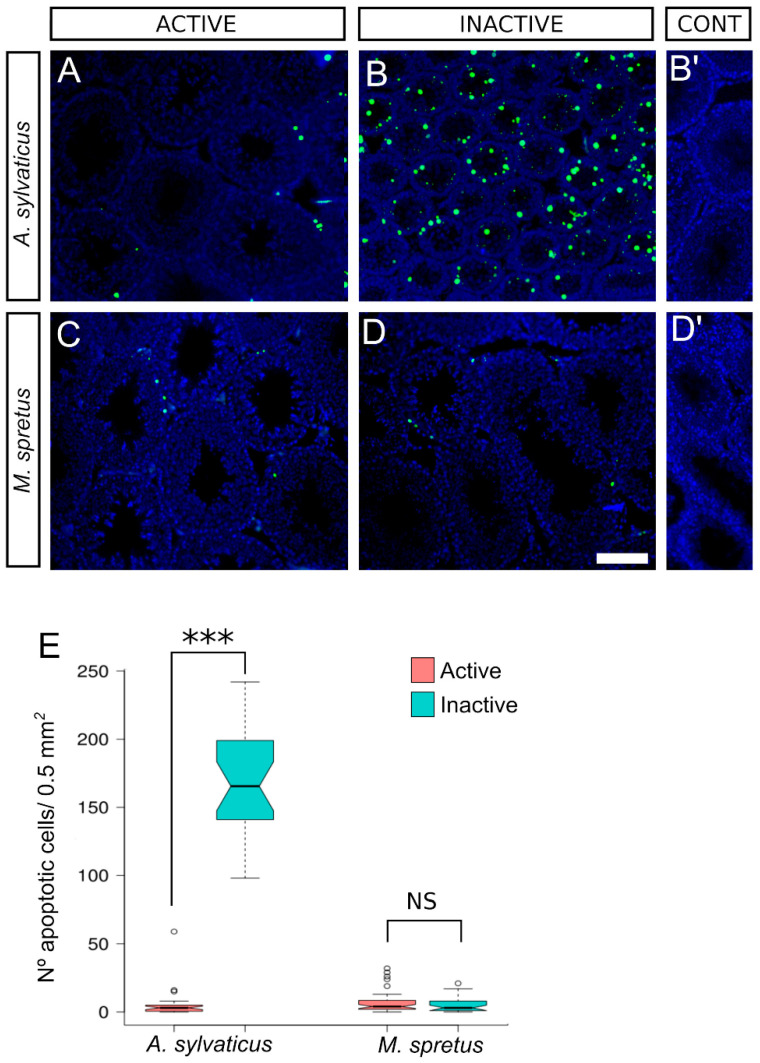
Increased levels of apoptosis in inactive testes of *A. sylvaticus*. (**A**–**D**) TUNEL assay in testis sections. In the active testes of *A. sylvaticus* (**A**) and in the active and inactive testes of *M. spretus* (**C**,**D**) most of the seminiferous tubules were devoid of apoptotic cells. In contrast, in inactive testes of *A. sylvaticus* were observed many apoptotic cells within the testis tubules (**B**). (**E**) Quantification of the number of apoptosis cell in active and inactive testes of *A. sylvaticus* and *M. spretus*. Testis sections in (**A**–**D**) were counterstained with DAPI. Inserts in (**B’**,**D’**) show negative controls. Scale bar shown in (**D**) represents 100 μm for (**A**–**D**). Asterisks indicate the level of statistical significance (***: *p* < 0.001); NS: not significant.

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
