# Peer review of "Divergent Seasonal Reproductive Patterns in Syntopic Populations of Two Murine Species in Southern Spain, Mus spretus and Apodemus sylvaticus"

_animals, 2021, doi:10.3390/ani11020243_

Round 1

Reviewer 1 Report

Manuscript ID: animals-1065967

Title: Divergent seasonal reproductive patterns in syntopic populations of two murine species in southern Spain, Mus spretus and Apodemus sylvaticus

Authors: Diaa Massoud, Miguel Lao-Pérez, Esperanza Ortega, Miguel Burgos, Rafael Jiménez *, Francisco J. Barrionuevo

Review

Seasonality of breeding is a widespread pattern of many mammals, and in the south non-breeding periods are not the same as in the north, even for the same species. Knowledge of the seasonality may have a practical importance, e.g., in species conservation or pest management. Therefore, reviewed manuscript is interesting and may attract readers.

Techniques of analyses and statistical treatment seems clear and well presented (but not properly referred). However, there are methodological drawbacks, which, until not cleared, do not warrant acceptance.

Main concern is about the sample of M. spretus. In the Line 115 is written “In addition, forty eight adult males”, however, Table S4 show average body mass of these being 12 g, and Table S2 – distribution of these values in the range of 8 (!) to 16 g.

We refer to Niethammer and Krapp Handbuch der Säugetiere Europas, where body mass of adult males of M. musculus spretus (the same species) is referred as 15.7 (12–18) g, and Palomo et al., 2009, where body mass of adult Spanish adult M. spretus is referred as 16.44 (15–21) g.

In the sample of current manuscript, body mass of adult males is obviously below these values; 62.5% of the sampled males are below 13 g, and 41.7% of sampled males are below 12 g.

Therefore, I see two possible solutions in this situation:

  1. Authors re-calculate their data using 18 individuals of M. spretus with body mass 13 g and over, and rewrite the text according new results, or
  2. They present unquestionable proof that males on M. spretus with body mass 8–12 g are really adult ones.

References cited:

Niethammer, J.; Krapp, F. (Eds.). Handbuch der Säugetiere Europas – Band 1, Nagetiere 1; AULA–Verlag: Wiesbaden, Germany, 1978, Table 99

Palomo, L. J., Justo, E. R., & Vargas, J. M. (2009). Mus spretus (Rodentia: muridae). Mammalian species, (840), 1-10.

Other comments

Lines 50-53: references needed

Lines 90-92: cited sources [21,22] did not match to text.

Line 99: wrong citation. In Amori, G., Gippoliti, S., & Helgen, K. M. (2008). Diversity, distribution, and conservation of endemic island rodents. Quaternary International, 182(1), 6–15. doi:10.1016/j.quaint.2007.05.014  M. spretus is not mentioned at all.

Line 172: mistype

Line 175: “gnu‐R free software (http://www.r‐project.org/)“ refers to nothing, the same as Windows – this is a shell for packages. Please refer properly,

Parts of Discussion nor are presented as Results (e.g., Lines 184-186; 211-213

Lines 186-187: see main comment above – part of the sample most possibly were not adult individuals.

Figure 1: wrong caption

Figure 2: wrong caption there are 2 scale bars in the figure

Lines 362-366: please check your text, Bronson stated a bit different things, namely “Food availability and ambient temperature determine energy balance, and variation in energy balance is the ultimate cause of seasonal breeding in all mammals and the proximate cause in many. Photoperiodic cueing is common among long-lived mammals from the highest latitudes down to the mid-tropics. It is much less common in shorter lived mammals at all latitudes.” I cannot agree with you.

Line 407: please check reference if they really show life expectancy?

Figure S1 – no scale presented

Author Response

We thank reviewer1 for his/her constructive criticism of our manuscript, corrections and suggestions that helped us to improve it.

Reviewer 1: Seasonality of breeding is a widespread pattern of many mammals, and in the south non-breeding periods are not the same as in the north, even for the same species. Knowledge of the seasonality may have a practical importance, e.g., in species conservation or pest management. Therefore, reviewed manuscript is interesting and may attract readers.

Techniques of analyses and statistical treatment seems clear and well presented (but not properly referred). However, there are methodological drawbacks, which, until not cleared, do not warrant acceptance.

Main concern is about the sample of M. spretus. In the Line 115 is written “In addition, forty eight adult males”, however, Table S4 show average body mass of these being 12 g, and Table S2 – distribution of these values in the range of 8 (!) to 16 g.

We refer to Niethammer and Krapp Handbuch der Säugetiere Europas, where body mass of adult males of M. musculus spretus (the same species) is referred as 15.7 (12–18) g, and Palomo et al., 2009, where body mass of adult Spanish adult M. spretus is referred as 16.44 (15–21) g.

In the sample of current manuscript, body mass of adult males is obviously below these values; 62.5% of the sampled males are below 13 g, and 41.7% of sampled males are below 12 g.

Therefore, I see two possible solutions in this situation:

  1. Authors re-calculate their data using 18 individuals of M. spretus with body mass 13 g and over, and rewrite the text according new results, or

  2. They present unquestionable proof that males on M. spretus with body mass 8–12 g are really adult ones.

References cited:

Niethammer, J.; Krapp, F. (Eds.). Handbuch der Säugetiere Europas – Band 1, Nagetiere 1; AULA–Verlag: Wiesbaden, Germany, 1978, Table 99

Palomo, L. J., Justo, E. R., & Vargas, J. M. (2009). Mus spretus (Rodentia: muridae). Mammalian species, (840), 1-10.

Response: Defining clear criteria to establish the age limits between juvenile and adult animals is very important in a study like this, so reviewer 1 concerns about this issue is reasonable. It is well known that mean body mass may vary among different populations of the same species and, as reviewer 1 noticed, the mean body mass of our adult males of M. spretus is obviously lower than that of other Iberian populations (only one animal in our sample weighted 17g and only two weighted 16g). Thus, the lower limit of adult body mass in the population we studied should not be established according to the values reported for other populations. In any case, the body mass was not the main criterion we used to determine if every particular male was adult or not. We used the testis mass (and the relative testis mass) and, mainly, the histology of the testes. Only animals showing active spermatogenesis were included in this study. These included a surprising male weighting just 8g but having active testes weighting 60mg. We also found males weighting 9, 10 and 11g with testes of 68, 96 and 97mg which were clearly adult ones, as the mass of their testes were near/above the mean testis mass of sexually active males in our sample (78.26±19.60 mg). So, we are confident that all males included in our initial sample were fertile with active spermatogenesis. Nevertheless, in order to prevent others from having the same doubts and concerns as reviewer 1, we have eliminated from the sample all potentially subadult males with body mass lower than 12g [in fact a 12g male had the biggest testes found in our study (111.3mg); see new Table S2] and have recalculated subsequent data in Table S4 and Figure 1. This change does not alter our conclusions at all. Also, we would like to make clear that all other analyses and experiments (protein immunodetection, BTB permeability, apoptosis incidence...) were performed using the testes from several males showing the highest body and testis mass in each period. A new sentence including this information has been inserted in the Material and Methods section.

Reviewer 1: Other comments

Lines 50-53: references needed

Response: A review is now cited there.

Reviewer 1: Lines 90-92: cited sources [21,22] did not match to text.

Response: The reviewer is right. There was a mistake during the automatic numbering of citations that passed unnoticed to us. We are sorry for this. The correct references are now included.

Reviewer 1: Line 99: wrong citation. In Amori, G., Gippoliti, S., & Helgen, K. M. (2008). Diversity, distribution, and conservation of endemic island rodents. Quaternary International, 182(1), 6–15. doi:10.1016/j.quaint.2007.05.014  M. spretus is not mentioned at all.

Response: The same as above. Correct references are now included

Reviewer 1: Line 172: mistype

Response: Corrected

Reviewer 1: Line 175: “gnu‐R free software (http://www.r‐project.org/)“ refers to nothing, the same as Windows – this is a shell for packages. Please refer properly,

Response: We made a search on how to cite informatic packages and found several articles indicating how to do it properly (for instance, see this one: https://easystats.github.io/report/articles/cite_packages.html). According to this, complete references are now included in the text.

Reviewer 1: Parts of Discussion nor are presented as Results (e.g., Lines 184-186; 211-213)

Response: The discussion sentences have been removed in the case of lines 184-186. However, the references to the expression of SOX9 in the testes of other species (lines 211-213) are there just to justify why we studied the expression of this gene also in A. sylvaticus and M. spretus.

Reviewer 1: Lines 186-187: see main comment above – part of the sample most possibly were not adult individuals.

Response: As stated above, the new calculations we did after removing the lightest males in both periods provide similar results as before.

Reviewer 1: Figure 1: wrong caption

Response: The names of species have been written in cursive font and the picture where the bar is placed (L) has been corrected.

Reviewer 1: Figure 2: wrong caption there are 2 scale bars in the figure

Response: We have corrected the names of the species, which are now written in cursive font. Regarding the scale bars, actually, there are three bars in Figure 2 as we included pictures made at three different magnifications. The caption includes references to all three bars as follows: Scale bar shown in D represents 50 μm for A-D. Scale bar shown in L represents 200 μm for I-L. Scale bar shown in P represents 100 μm for E-H and M-P”. Thus, we think everything is correct now.

Reviewer 1: Lines 362-366: please check your text, Bronson stated a bit different things, namely “Food availability and ambient temperature determine energy balance, and variation in energy balance is the ultimate cause of seasonal breeding in all mammals and the proximate cause in many. Photoperiodic cueing is common among long-lived mammals from the highest latitudes down to the mid-tropics. It is much less common in shorter lived mammals at all latitudes.” I cannot agree with you.

Response: The sentence has been modified accordingly.

Reviewer 1: Line 407: please check reference if they really show life expectancy?

Response: The reviewer is right. The correct reference is now included

Reviewer 1: Figure S1 – no scale presented

Response: A scale bar has been added to Figure S1.

Reviewer 2 Report

The study by Massound et al., reports for the first time comparison analysis on testis biology of two Spanish seasonal rodent species Mus spretus and Apodemus sylvaticus. Before publishing article needs revision in aspects listed below.

  1. lines 59-63 reports on bank vole reproduction should be cited. As it is a unique but well-described model of seasonal reproduction
  2. Give age of animals. Adult can be also ageing.
  3. Give details on controls for ihc analyses.
  4. lines 57-59 give citations and more details. “hormonal variations”? rather hormone production or levels
  5. lines 104-107 provide more details what will be studied. Study aim should be more specified.  In addition, you study biology of seasonal testis as you use both actively reproducing and regressed animals.
  6. line 104 “neither” is not needed
  7. line 194 “whereas” is not needed
  8. In M and M there is no information that epididymis will be also studied. It should be clarified. Information throughout the manuscript text on what part of epididymis is exactly study is needed.
  9. Results and Discussion should be separated e.g. line 215 move what will more fits into Discussion
  10. For studies of blood-testis barrier and communication between seminiferous epithelium cells connexin 43 staining should be used. In addition, CX43 presence would give information on interstitium biology. In addition, Leydig cells should be studies by 3beta-HSD staining. AR staining should be not linking with steroidogenesis.
  11. It is a pity that quantitative analyses e.g. western blot for clear information was not performed here.

Author Response

We thank reviewer2 for her/his constructive criticism of our manuscript, corrections and suggestions that helped us to improve it.

The study by Massound et al., reports for the first time comparison analysis on testis biology of two Spanish seasonal rodent species Mus spretus and Apodemus sylvaticus. Before publishing article needs revision in aspects listed below.

Reviewer 2: lines 59-63 reports on bank vole reproduction should be cited. As it is a unique but well-described model of seasonal reproduction

Response: The case of the bank vole has been included and three references have been cited

Reviewer 2: Give age of animals. Adult can be also ageing.

Response: The main goal of this work was to establish the differences between adult males of two species in two well defined periods of their circannual reproductive cycle. Thus, our main requirement was to make sure that males included in the study were in fact fertile adult ones. For this, we used body and testis mass and histological data, as mentioned above, which provide highly reliable results for our purposes. Determining of the absolute age of every individual in this study would have been very time consuming and not very useful for two reasons: 1) available methods for age determination in feral mammalian specimens (dental wear, dry lens mass, tooth histology) are laborious and not very accurate in small rodents with a very short life expectancy; 2) because of their short life expectancy, oldest males also retain active spermatogenesis. In fact, the heaviest males we analyzed had fully functional testes. Thus, knowing their absolute age would not have been really useful for our study.

Reviewer 2: Give details on controls for ihc analyses.

Response: controls were preparations in which the primary antibody was omitted. This was indicated in the Material and Methods section of the first version of the manuscript (lines 137-138): The primary antibody was omitted in negative controls.”

Reviewer 2: lines 57-59 give citations and more details. “hormonal variations”? rather hormone production or levels

Response: The text has been changed and four new references were included.

Reviewer 2: lines 104-107 provide more details what will be studied. Study aim should be more specified. In addition, you study biology of seasonal testis as you use both actively reproducing and regressed animals.

Response: It is not clear for us what the reviewer means with these comments, as all his/her requests are already included in our original text:

1) the reviewer asks us toprovide more details what will be studied. Study aim should be more specified”; we stated that our aim is to study the process of seasonal testis regression in two species (the just above mentioned A. sylvaticus and M. sprestus) in which no such study had previously been performed. Then, to be more specific, we stated that we performed a morphometric, histological, hormonal, and gene-expression study of the testes in populations of these two species. 2) Then, the reviewer states “In addition, you study biology of seasonal testis as you use both actively reproducing and regressed animals”, when our paragraph finishes by the way indicating that we have studied these animals during the two periods of their respective reproductive cycles in south-eastern Iberian peninsula.

We lightly modified this last sentence, but consider that this last paragraph of the Introduction section should not anticipate much more information that will be detailed later in the text.

Reviewer 2: line 104 “neither” is not needed

Response: “neither of” was deleted

Reviewer 2: line 194 “whereas” is not needed

Response: “whereas” is needed here to emphasize the contrast between the situations in the two species

Reviewer 2: In M and M there is no information that epididymis will be also studied. It should be clarified. Information throughout the manuscript text on what part of epididymis is exactly study is needed.

Response: In lines 126-127 of the original version of the manuscript we wrote: “Epididymides were also processed for further histological studies”. Now we have added the following sentence: The caput region was usually analyzed”

Reviewer 2: Results and Discussion should be separated e.g. line 215 move what will more fits into Discussion

Response: As indicated above, the references to the expression of SOX9 in the testes of other species (lines 211-213) are there just to justify why we studied the expression of this gene also in A. sylvaticus and M. Spretus. It is not an interpretation of results.

Reviewer 2: For studies of blood-testis barrier and communication between seminiferous epithelium cells connexin 43 staining should be used. In addition, CX43 presence would give information on interstitium biology. In addition, Leydig cells should be studies by 3beta-HSD staining. AR staining should be not linking with steroidogenesis.

Response: Reviewer is right suggesting that many other cell markers could have been used in this study. In fact, we tried several additional antibodies, including conexin 43, but they did not worked properly in the studied species (this is a common drawback when working with non-model species). AR staining is not involved in steroidogenesis, but this protein is the main intracellular mediator of androgen action and this is the reason why we included it into this subsection of the Results section.

Reviewer 2: It is a pity that quantitative analyses e.g. western blot for clear information was not performed here.

Response: The reviewer is right indicating that quantitative analyses of the expression profiles of the genes investigated could have added some value to this study. In this respect, we would like to say that, even without additional experiments, our study implied several years work both in the field and in the lab. We considered the possibility of including this kind of analyses when we begun to study testis regression in wild small mammals, but we desisted for several reasons. Since many of the testes we investigated were really tiny, we could not be able to get protein extracts large enough to perform western blots, so we would have to use qRT-PCR. In this case, considering that we are dealing with feral, non-model species, specific gene fragments should be cloned and sequenced in order to design appropriate primers for each species and gene. Also, we consider that quantitative studies are not indispensable here as we established only two study groups (sexually active and inactive) and thus gene expression analysis could be done in terms of presence or absence of the corresponding protein, for which IHC and IF are suitable methods. In conclusion, knowing the precise expression levels of the genes would provide little additional information, which does not justify the excessive work load it represents.

Round 2

Reviewer 1 Report

Manuscript ID: animals-1065967

Title: Divergent seasonal reproductive patterns in syntopic populations of two murine species in southern Spain, Mus spretus and Apodemus sylvaticus

Authors: Diaa Massoud, Miguel Lao-Pérez, Esperanza Ortega, Miguel Burgos, Rafael Jiménez *, Francisco J. Barrionuevo

Review, round 2

While I am satisfied with the changes in manuscript (and mainly, for removing smallest individuals from statistics in M. spretus), presented text has many typing and formatting mistakes. I suggest consulting with the Requirements and published papers. Perhaps, authors need more time to read it make necessary changes:

Throughout the text: use long dash between reference numbers, i.e, [2–5], not [2-5], Line 131  [4–6], not [4-6], etc.

Line 55: males, not male

[7] is not on C. glareolus

Line 176  0.55, not 0,55

Lines 510-515 formatting

Line 580, [20], [27] wrong doi presentation

[17, 18], and other references: do not capitalize second word in species name, auratus, not Auratus, line 594 californicus, not Californicus

[26] missing part of the source

[63] – formatting

[39] and other references – not all journal names abbreviated properly

Tables S4, S5 – add explanation as for significance levels, denoted by asterisks

Author Response

We thank reviewer 1 for his/her careful reading of the manuscript and their constructive remarks.

While I am satisfied with the changes in manuscript (and mainly, for removing smallest individuals from statistics in M. spretus), presented text has many typing and formatting mistakes. I suggest consulting with the Requirements and published papers. Perhaps, authors need more time to read it make necessary changes:

Reviewer 1: Throughout the text: use long dash between reference numbers, i.e, [2–5], not [2-5], Line 131 [4–6], not [4-6], etc.

Response: All similar dashes have been changed

Reviewer 1: Line 55: males, not male

Response: Corrected

Reviewer 1: [7] is not on C. glareolus

Response: On request of reviewer 2, references 6-8 have been substituted by three different ones.

Reviewer 1: Line 176 0.55, not 0,55

Response: Corrected

Reviewer 1: Lines 510-515 formatting

Response: Corrected

Reviewer 1: Line 580, [20], [27] wrong doi presentation

Response: the DOI links have been erased in refs 20 and 27, as well as in many other references

Reviewer 1: [17, 18], and other references: do not capitalize second word in species name, auratus, not Auratus, line 594 californicus, not Californicus

Response: This mistake has been corrected in all references with names of species. We used the “Animalsstyle of the Zotero application to generate the reference list. So, the mistake of capitalizing the specific names (and adding DOI links) was produced during this automatic process. Nevertheless, we acknowledge that the generated mistakes passed unnoticed to us, but perhaps the editors of the journal should inform about this problem.

Reviewer 1: [26] missing part of the source

Response: The missing part has been added (Zotero)

Reviewer 1: [63] – formatting

Response: Corrected (Zotero)

Reviewer 1: [39] and other references – not all journal names abbreviated properly

Response: the journal names have been properly abbreviated in all references (Zotero)

Reviewer 1: Tables S4, S5 – add explanation as for significance levels, denoted by asterisks

Response: the requested explanations have been added

Reviewer 2 Report

I am not fully satisfied with the correction. Below there are some information on which points should be furtherly improved.

The case of the bank vole has been included and three references have been cited
More recent data should be cited e.g. Profaska-Szymik et al., IJMS 2020; Milon et al., Gen Comp Endocrinol 2019; Pawlicki et al., J Physiol Pharmacol 2017

controls were preparations in which the primary antibody was omitted. This was indicated in the Material and Methods section of the first version of the manuscript (lines 137-138): “The primary antibody was omitted in negative controls.”
Could you add negative controls to the ihc Figures?

lines 104-107 provide more details what will be studied. It is not clear for us what the reviewer means with these comments, as all his/her requests are already included in our original text
In the aim of the study there is not detailed information presented that will be studied, androgen action (AR expression) and/or level, BTB etc. will be studied. Please consider adding that information. In addition our study is comparative what is also important to highlight.

In lines 126-127 of the original version of the manuscript we wrote: “Epididymides were also processed for further histological studies”. Now we have added the following sentence: “The caput region was usually analyzed”
That is true but please check once again through the whole M&M text for example in histological analysis you only inform that you use testicular sections…

AR staining is not involved in steroidogenesis, but this protein is the main intracellular mediator of androgen action and this is the reason why we included it into this subsection of the Results section. If the authors agreed with the Reviewer than the title of the section should be
Androgen signaling and level is reduced in the inactive testes of both A. sylvaticus and M. spretussylvaticus and M. spretus

Author Response

We thank Reviewer2 for his/her careful reading of the manuscript and their constructive comments.

Reviewer 2: I am not fully satisfied with the correction. Below there are some information on which points should be furtherly improved.

The case of the bank vole has been included and three references have been cited
More recent data should be cited e.g. Profaska-Szymik et al., IJMS 2020; Milon et al., Gen Comp Endocrinol 2019; Pawlicki et al., J Physiol Pharmacol 2017

Response: The suggested references have been included and the previous ones have been removed.

Reviewer 2: controls were preparations in which the primary antibody was omitted. This was indicated in the Material and Methods section of the first version of the manuscript (lines 137-138): “The primary antibody was omitted in negative controls.” Could you add negative controls to the ihc Figures?

Response: Inserts containing images of every negative control have been included in the figures.

Reviewer 2: lines 104-107 provide more details what will be studied. It is not clear for us what the reviewer means with these comments, as all his/her requests are already included in our original text
In the aim of the study there is not detailed information presented that will be studied, androgen action (AR expression) and/or level, BTB etc. will be studied. Please consider adding that information. In addition our study is comparative what is also important to highlight.

Response: The requested additional information has been added to this paragraph.

Reviewer 2: In lines 126-127 of the original version of the manuscript we wrote: “Epididymides were also processed for further histological studies”. Now we have added the following sentence: “The caput region was usually analyzed”
That is true but please check once again through the whole M&M text for example in histological analysis you only inform that you use testicular sections…

Response: The 2.2. subsection of the Material and Methods section (Histology and immunohistological methods) has been modified to make clear that epididymides were only processed for haematoxilin-eosin staining and that immunodetection methods were applied only to testis sections.

Reviewer 2: AR staining is not involved in steroidogenesis, but this protein is the main intracellular mediator of androgen action and this is the reason why we included it into this subsection of the Results section. If the authors agreed with the Reviewer than the title of the section should be Androgen signaling and level is reduced in the inactive testes of both A. sylvaticus and M. spretus

Response: We have modified the title of this subsection as follows: “Androgen levels are reduced in the inactive males of both A. sylvaticus and M. Spretus”. We found no differences between the two species regarding AR expression.